# Visualizing the PHATE of Neural Networks

**Scott Gigante**
Comp. Biol. and Bioinf. Program
Yale University
New Haven, CT 06511
scott.gigante@yale.edu

**Adam S. Charles**
Princeton Neuroscience Institute
Princeton University
Princeton, NJ, 08544
adamsc@princeton.edu

**Smita Krishnaswamy**
Depts. of Genetics and Computer Science
Yale University
New Haven, CT 06520
smita.krishnaswamy@yale.edu

**Gal Mishne**
Halıcıoğlu Data Science Institute
University of California, San Diego
La Jolla, CA 92093
gmishne@ucsd.edu

## Abstract

Understanding why and how certain neural networks outperform others is key to guiding future development of network architectures and optimization methods. To this end, we introduce a novel visualization algorithm that reveals the internal geometry of such networks: Multislice PHATE (M-PHATE), the first method designed explicitly to visualize how a neural network's hidden representations of data evolve throughout the course of training. We demonstrate that our visualization provides intuitive, detailed summaries of the learning dynamics beyond simple global measures (i.e., validation loss and accuracy), without the need to access validation data. Furthermore, M-PHATE better captures both the dynamics and community structure of the hidden units as compared to visualization based on standard dimensionality reduction methods (e.g., ISOMAP, t-SNE). We demonstrate M-PHATE with two vignettes: continual learning and generalization. In the former, the M-PHATE visualizations display the mechanism of "catastrophic forgetting" which is a major challenge for learning in task-switching contexts. In the latter, our visualizations reveal how increased heterogeneity among hidden units correlates with improved generalization performance. An implementation of M-PHATE, along with scripts to reproduce the figures in this paper, is available at https://github.com/scottgigante/M-PHATE.

## 1 Introduction

Despite their massive increase in popularity in recent years, deep networks are still regarded as opaque and difficult to interpret or analyze. Understanding how and why certain neural networks perform better than others remains an art. The design of neural networks and their training: choice of architectures, regularization, activation functions, and hyperparameters, while informed by theory and prior work, is often driven by intuition and tuned manually [1]. The combination of these intuition-driven selections and long training times even on high-performance hardware (e.g., 3 weeks on 8 GPUs for the popular ResNet-200 network for image classification), means that the combinatorial task of testing all possible choices is impossible, and must be guided by more principled evaluations and explorations.

A natural and widely used measure of evaluation for the difference between network architectures and optimizers is the validation loss. In some situations, the validation loss lacks a clearly defined global meaning, i.e., when the loss function itself is learned, and other evaluations are required [2, 3]. While

such scores are useful for ranking models on the basis of performance, they crucially do not explain why one model outperforms another. To provide additional insight, visualization tools have been employed, for example to analyze the "loss landscape" of a network. Specifically, these visualizations depict how architectural choices modify the smoothness of local minima [4, 5] — a quality assumed to be related to generalization abilities.

Local minima smoothness, however, is only one possible correlate of performance. Another internal quality that can be quantified is the hidden representations of inputs provided by the hidden unit activations. The multi-layered hidden representations of data are, in effect, the single most important feature distinguishing neural networks from classical machine learning techniques in generalization [6–10]. We can view the changes in representation by stochastic gradient descent as a dynamical system evolving from its random initialization to a converged low-energy state. Observing the progression of this dynamical system gives more insight into the learning process than simply observing it at a single point in time (e.g., after convergence.) In this paper, we contribute a novel method of inspecting a neural network's learning: we visualize the evolution of the network's hidden representation during training to isolate key qualities predictive of improved network performance.

Analyzing extremely high-dimensional objects such as deep neural networks requires methods that can reduce these large structures into more manageable representations that are efficient to manipulate and visualize. Dimensionality reduction is a class of machine learning techniques which aim to reduce the number of variables under consideration in high-dimensional data while maintaining the structure of a dataset. There exist a wide array of dimensionality reduction techniques designed specifically for visualization, which aim specifically to capture the structure of a dataset in two or three dimensions for the purposes of human interpretation, e.g., MDS [11], t-SNE [12], and Isomap [13]. In this paper, we employ PHATE [14], a kernel-based dimensionality reduction method designed specifically for visualization which uses multidimensional scaling (MDS) [11] to effectively embed the diffusion geometry [15] of a dataset in two or three dimensions.

In order to visualize the evolution of the network's hidden representation, we take advantage of the longitudinal nature of the data; we have in effect many observations of an evolving dynamical system, which lends itself well to building a graph from the data connecting observations across different points in time. We construct a weighted multislice graph (where a "slice" refers to the network state at a fixed point in time) by creating connections between hidden representations obtained from a single unit across multiple epochs, and from multiple units within the same epoch. A pairwise affinity kernel on this graph reflects the similarity between hidden units and their evolution over time. This kernel is then dimensionality reduced with PHATE and visualized in two dimensions.

The main contributions of this paper are as follows. We present Multislice PHATE (M-PHATE), which combines a novel multislice kernel construction with the PHATE visualization [14]. Our kernel captures the dynamics of an evolving graph structure, that when when visualized, gives unique intuition about the evolution of a neural network over the course of training and re-training. We compare M-PHATE to other dimensionality reduction techniques, showing that the combined construction of the multislice kernel and the use of PHATE provide significant improvements to visualization. In two vignettes, we demonstrate the use M-PHATE on established training tasks and learning methods in continual learning, and in regularization techniques commonly used to improve generalization performance. These examples draw insight into the reasons certain methods and architectures outperform others, and demonstrate how visualizing the hidden units of a network with M-PHATE provides additional information to a deep learning practitioner over classical metrics such as validation loss and accuracy, all without the need to access validation data.

## 2  Background

Diffusion maps (DMs) [15] is an important nonlinear dimensionality reduction method that has been used to extract complex relationships between high-dimensional data  [16–22]. PHATE [14] aims to optimize diffusion maps for data visualization. We briefly review the two approaches.

Given a high-dimensional dataset $\{x_i\}$, DMs operate on a pairwise similarity matrix $\mathbf{W}$ (e.g., computed via a Gaussian kernel $\mathbf{W}(x_i, x_j) = \exp\{-\|x_i - x_j\|^2/\epsilon\}$). and return an embedding of the data in a low-dimensional Euclidean space. To compute this embedding, the rows of $\mathbf{W}$ are normalized by $\mathbf{P} = \mathbf{D}^{-1}\mathbf{W}$, where $\mathbf{D}_{ii} = \sum_j \mathbf{W}_{ij}$. The resulting matrix $\mathbf{P}$ can be interpreted as the transition matrix of a Markov chain over the dataset and powers of the matrix, $\mathbf{P}^t$, represents running

the Markov chain forward $t$ steps. The matrix $\mathbf{P}$ thus has a complete sequence of bi-orthogonal left and right eigenvectors $\phi_i$, $\psi_i$, respectively, and a corresponding sequence of eigenvalues $1 = \lambda_0 \geq |\lambda_1| \geq |\lambda_2| \geq \ldots$. Due to the fast spectrum decay of $\{\lambda_l\}$, we can obtain a low-dimensional representation of the data using only the top $\ell$ eigenvectors. Diffusion maps, defined as $\Psi_t(x) = (\lambda_1^t \psi_1(x), \lambda_2^t \psi_2(x), \ldots, \lambda_\ell^t \psi_\ell(x))$, embeds the data points into a Euclidean space $\mathbb{R}^\ell$ where the Euclidean distance approximates the diffusion distance:

$$\mathbf{D}_t^2(x_i, x_j) = \sum_{x_k} \frac{(p_t(x_i, x_k) - p_t(x_j, x_k))^2}{\phi_0(x_j)} \approx \| \Psi_t(x_i) - \Psi_t(x_j) \|_2^2$$

Note that $\psi_0$ is neglected because it is a constant vector.

To enable successful data visualization, a method must reduce the dimensionality to two or three dimensions; diffusion maps, however, reduces only to the intrinsic dimensionality of the data, which may be much higher. Thus, to calculate a 2D or 3D representation of the data, PHATE applies MDS [11] to the *informational distance* between rows $i$ and $j$ of the diffusion kernel $\mathbf{P}^t$ defined as

$$\mathbf{\Phi}_t(i, j) = \| \log \mathbf{P}^t(i) - \log \mathbf{P}^t(j) \|_2$$

where $t$ is selected automatically as the knee point of the Von Neumann Entropy of the diffusion operator. For further details, see Moon et al. [14].

## 2.1 Related work

We consider the evolving state of a neural network's hidden units as a dynamical system which can be represented as a *multislice graph* on which we construct a pairwise affinity kernel. Such a kernel considers both similarities between hidden units in the same epoch or time-slice (denoted *intraslice* similarities) and similarities of a hidden unit to itself across different time-slices (denoted *interslice* similarities). The concept of constructing a graph for data changing over time is motivated by prior work both in harmonic analysis [20, 23–25, 22] and network science [26]. For example, Coifman and Hirn [20] suggest an algorithm for jointly analyzing DMs built over data points that are changing over time by aligning the separately constructed DMs, while Mucha et al. [26] suggest an algorithm for community detection in multislice networks by connecting each node in one network slice to itself in other slices, with identical *fixed weights* for all intraslice connections. In both cases, such techniques are designed to detect changes in intraslice dynamics over time, yet interslice dynamics are not incorporated into the model.

## 3 Multiscale PHATE

### 3.1 Preliminaries

Let $F$ be a neural network with a total of $m$ hidden units applied to $d$-dimensional input data. Let $F_i : \mathbb{R}^d \to \mathbb{R}$ be the activation of the $i$th hidden unit of $F$, and $F^{(\tau)}$ be the representation of the network after being trained for $\tau \in \{1, \ldots, n\}$ epochs on training data $X$ sampled from a dataset $\mathcal{X}$.

A natural feature space for the hidden units of $F$ is the activations of the units with respect to the input data. Let $Y \subset \mathcal{X}$ be a representative sample of $p \ll |X|$ points. (In this paper, we use points not used in training; however, this is not necessary. Further discussion of this is given in Section S2.) Let $Y_k$ be the $k$th sample in $Y$. We use the hidden unit activations $F(Y)$ to compute a shared feature space of dimension $p$ for the hidden units. We can then calculate similarities between units from all layers. Note that one may instead consider the hidden units' learned parameters (e.g. weight matrices and bias terms); however, these are not suitable for our purposes as they are not necessarily the same shape between hidden layers, and additionally the parameters may contain information not relevant to the data (for example, in dimensions of $\mathcal{X}$ containing no relevant information.)

We denote the *time trace* $\mathbf{T}$ of the network as a $n \times m \times p$ tensor containing the activations at each epoch $\tau$ of each hidden unit $F_i$ with respect to each sample $Y_k \in Y$. We note that in practice, the major driver of variation in $\mathbf{T}$ is the bias term contributing a fixed value to the activation of each hidden unit. Further, we note that the absolute values of the differences in activation of a hidden unit are not strictly meaningful, since any differences in activation can simply be magnified by a larger

kernel weight in the following layer. Therefore, to calculate more meaningful similarities, we first $z$-score the activations of each hidden unit at each epoch $\tau$

$$\mathbf{T}(\tau, i, k) = \frac{F_i^{(\tau)}(Y_k) - \frac{1}{p}\sum_\ell F_i^{(\tau)}(Y_\ell)}{\sqrt{\mathrm{Var}_\ell\, F_i^{(\tau)}(Y_\ell)}}.$$

## 3.2 Multislice Kernel

The time trace gives us a natural substrate from which to construct a visualization of the network's evolution. We construct a kernel over $\mathbf{T}$ utilizing our prior knowledge of the temporal aspect of $\mathbf{T}$ to capture its dynamics. Let $\mathbf{K}$ be a $nm \times nm$ kernel matrix between all hidden units at all epochs (the $(\tau m + j)$th row or column of $K$ refers to $j$-th unit at epoch $\tau$).We henceforth refer to the $(\tau m + j)$th row of $\mathbf{K}$ as $\mathbf{K}((\tau, j), :)$ and the $(\tau m + j)$th column of $\mathbf{K}$ as $\mathbf{K}(:, (\tau, j))$.

To capture both the evolution of a hidden unit throughout training as well as its community structure with respect to other hidden units, we construct a multislice kernel matrix which reflects both affinities between hidden units $i$ and $j$ in the same epoch $\tau$, or intraslice affinities

$$\mathbf{K}_{\text{intraslice}}^{(\tau)}(i, j) = \exp\left(-\|\mathbf{T}(\tau, i) - \mathbf{T}(\tau, j)\|_2^\alpha / \sigma_{(\tau, i)}^\alpha\right)$$

as well as affinities between a hidden unit $i$ and itself at different epochs, or interslice affinities

$$\mathbf{K}_{\text{interslice}}^{(i)}(\tau, \upsilon) = \exp\left(-\|\mathbf{T}(\tau, i) - \mathbf{T}(\upsilon, i)\|_2^2 / \epsilon^2\right)$$

where $\sigma_{(\tau, i)}$ is the intraslice bandwidth for unit $i$ at epoch $\tau$, $\epsilon$ is the fixed intraslice bandwidth, and $\alpha$ is the adaptive bandwidth decay parameter.

In order to maintain connectivity while increasing robustness to parameter selection for the intraslice affinities $\mathbf{K}_{\text{intraslice}}^{(\tau)}$, we use an adaptive-bandwidth Gaussian kernel (termed the *alpha-decay kernel* [14]), with bandwidth $\sigma_{(\tau, i)}$ set to be the distance of unit $i$ at epoch $\tau$ to its $k$th nearest neighbor across units at that epoch: $\sigma_{(\tau, i)} = d_k(\mathbf{T}(\tau, i), \mathbf{T}(\tau, :))$, where $d_k(x, X)$ denotes the $L_2$ distance from $x$ to its $k$th nearest neighbor in $X$. Note that the use of the adaptive bandwidth means that the kernel is not symmetric and will require symmetrization. In order to allow the kernel to represent changing dynamics of units over the course of learning, we use a fixed-bandwidth Gaussian kernel in the interslice affinities $\mathbf{K}_{\text{interslice}}^{(i)}$, where $\epsilon$ is the average across all epochs and all units of the distance of unit $i$ at epoch $\tau$ to its $\kappa$th nearest neighbor among the set consisting of the same unit $i$ at all other epochs $\epsilon = \frac{1}{nm}\sum_{\tau=1}^{n}\sum_{i=1}^{m} d_\kappa(\mathbf{T}(\tau, i), \mathbf{T}(:, i))$.

Finally, the multislice kernel matrix contains one row and column for each unit at each epoch, such that the intraslice affinities form a block diagonal matrix and the interslice affinities form off-diagonal blocks composed of diagonal matrices (see Figures S1 and S2 for a diagram):

$$\mathbf{K}((\tau, i), (\upsilon, j)) = \begin{cases} \mathbf{K}_{\text{intraslice}}^{(\tau)}(i, j), & \text{if } \tau = \upsilon; \\ \mathbf{K}_{\text{intraslice}}^{(i)}(\tau, \upsilon), & \text{if } i = j; \\ 0, & \text{otherwise.} \end{cases}$$

We symmetrize this kernel as $\mathbf{K}' = \frac{1}{2}(\mathbf{K} + \mathbf{K}^T)$, and row normalize it to obtain $\mathbf{P} = \mathbf{D}^{-1}\mathbf{K}$, which represents a random walk over all units across all epochs, where propagating from $(\tau, i)$ to $(\nu, j)$ is conditional on the transition probabilities between epochs $\tau$ and $\nu$. PHATE [14] is applied to $\mathbf{P}$ to visualize the time trace $\mathbf{T}$ in two or three dimensions.

# 4 Results

## 4.1 Example visualization

To demonstrate our visualization, we train a feedforward neural network with 3 layers of 64 hidden units to classify digits in MNIST [27]. The visualization is built on the time trace $T$ evaluated on the network over a single round of training that lasted 300 epochs and reached 96% validation accuracy.

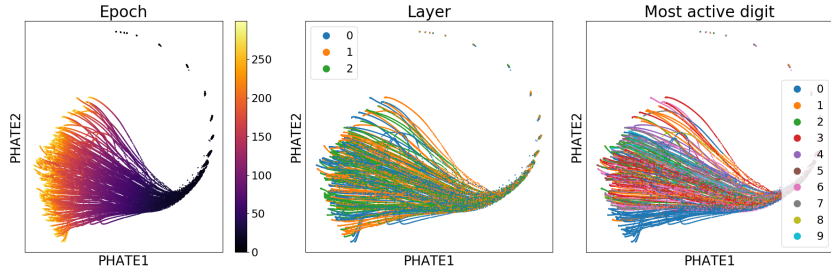

Figure 1: Visualization of a simple 3-layer MLP trained on MNIST with M-PHATE. Visualization is colored by epoch (left), hidden layer (center), and most active digit for each unit (right).

We visualize the network using M-PHATE (Fig. 1) colored by epoch, hidden layer and the digit for which examples of that digit most strongly activate the hidden unit. The embedding is clearly organized longitudinally by epoch, with larger jumps between early epochs and gradually smaller steps as the network converges. Additionally, increased structure emerges in the latter epochs as the network learns meaningful representations of the digits, and groups of neurons activating on the same digits begin to co-localize. Neurons of different layers frequently co-localize, showing that our visualization allows meaningful comparison of hidden units in different hidden layers.

## 4.2 Comparison to other visualization methods

To evaluate the quality of the M-PHATE visualization, we compare to three established visualization methods: diffusion maps, t-SNE and ISOMAP. We also compare our multislice kernel to the standard formalism of these visualization techniques, by computing pairwise distances or affinities between all units at all time points without taking into account the multislice nature of the data.

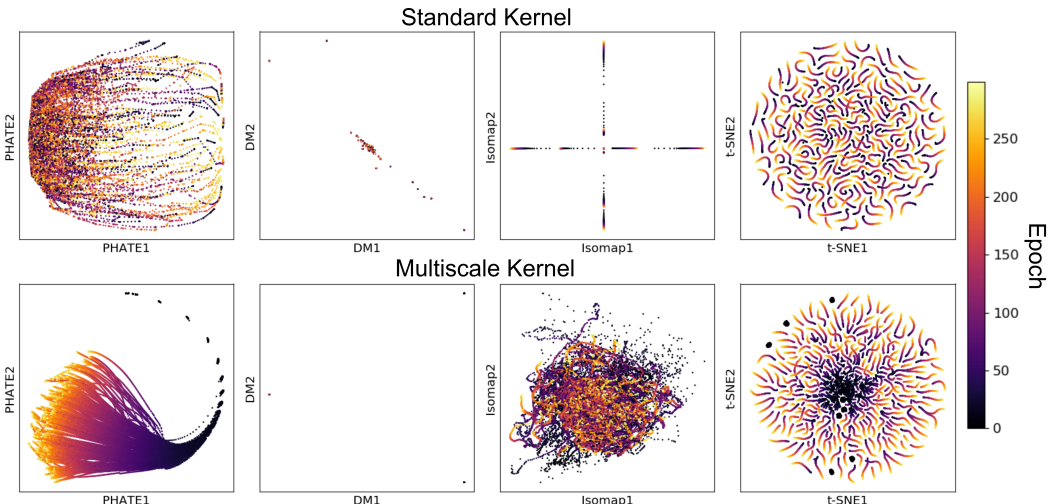

Figure 2: Comparison of standard application of visualization algorithms. Each point represents a hidden unit at a given epoch during training and is colored by the epoch.

Figure 2 shows the standard and multislice visualizations for all four dimensionality reduction techniques of the network in Section 4.1. For implementation details, see Section S3. Only the Multislice PHATE visualization reveals any meaningful evolution of the neural network over time. To quantify the quality of the visualization, we compare both interslice and intraslice neighborhoods in the embedding to the equivalent neighborhoods in the original data. Specifically, for a visualization $V$ we define the intraslice neighborhood preservation of a point $V(t, i) \in V$ as

$$\frac{1}{|k|} \left| \mathcal{N}^k_{V(\tau,:)}(V(\tau, i)) \cap \mathcal{N}^k_{T(\tau,:)}(T(\tau, i)) \right|$$

Table 1: Neighborhood preservation of visualization methods applied to a FFNN classifying MNIST.

| | Multislice | | | | Standard | | | |
| --- | --- | --- | --- | --- | --- | --- | --- | --- |
| | PHATE | DM | Isomap | t-SNE | PHATE | DM | Isomap | t-SNE |
| Intraslice, $k = 10$ | **0.26** | 0.19 | 0.11 | 0.13 | 0.05 | 0.09 | 0.06 | 0.06 |
| Interslice, $k = 10$ | 0.95 | 0.58 | 0.79 | 0.91 | 0.47 | 0.44 | 0.68 | **0.96** |
| Intraslice, $k = 40$ | **0.45** | 0.36 | 0.25 | 0.26 | 0.21 | 0.26 | 0.22 | 0.22 |
| Interslice, $k = 40$ | 0.93 | 0.75 | 0.78 | 0.92 | 0.67 | 0.54 | 0.70 | **0.94** |
| Loss Correlation | **0.81** | 0.61 | 0.61 | 0.33 | 0.25 | 0.13 | 0.47 | -0.04 |

and the interslice neighborhood preservation of $V(t, i)$ as

$$\frac{1}{|k|} \left| \mathcal{N}_{V(:,i)}^k (V(\tau, j)) \cap \mathcal{N}_{T(:,i)}^k (T(\tau, i)) \right|$$

where $\mathcal{N}_X^k(x)$ denotes the $k$ nearest neighbors of $x$ in $X$. We also calculate the Spearman correlation of the rate of change of each hidden unit with the rate of change of the validation loss to quantify the fidelity of the visualization to the diminishing rate of convergence towards the end of training.

M-PHATE achieves the best neighborhood preservation on all measures except the interslice neighborhood preservation, in which it performs on-par with standard t-SNE. Additionally, the multislice kernel construction outperforms the corresponding standard kernel construction for all methods and all measures, except again in the case of t-SNE for interslice neighborhood preservation. M-PHATE also has the highest correlation with change in loss, making it the most faithful display of network convergence.

## 4.3 Continual learning

An ongoing challenge in artificial intelligence is in making a single model perform well on many tasks independently. The capacity to succeed at dynamically changing tasks is often considered a hallmark of genuine intelligence, and is thus crucial to develop in artificial intelligence [28]. Continual learning is one attempt at achieving this goal sequentially training a single network on different tasks with the aim of instilling the network with new abilities as data becomes available.

To assess networks designed for continual learning tasks, a set of training baselines have been proposed. Hsu et al. [29] define three types of continual learning scenarios for classification: incremental task learning, in which a separate binary output layer is used for each task; incremental domain learning, in which a single binary output layer performs all tasks; and incremental class learning, in which a single 10-unit output layer is used, with each pair of output units used for just a single task. Further details are given in Section S4.

We implemented a 2-layer MLP with 400 units in each hidden layer to perform incremental, domain and class learning tasks using three described baselines: standard training with Adagrad [30] and Adam [31], and an experience replay training scheme called Naive Rehearsal [29] in which a small set of training examples from each task are retained and replayed to the network during subsequent tasks. Each network was trained for 4 epochs before switching to the next task. Overall, we find that validation performance is fairly consistent with results reported in Hsu et al. [29], with Naive Rehearsal performing best, followed by Adagrad and Adam. Class learning was the most challenging, followed by domain learning and task learning.

Figure 3 shows M-PHATE visualizations of learning in networks trained in each of three baselines, with network slices taken every 50 batches rather than every epoch for increased resolution. Notably, we observe a stark difference in how structure is preserved over training between networks, which is predictive of task performance. The highest-performing networks all tend to preserve representational structure across changing tasks. On the other hand, networks trained with Adam — the worst performing combinations — tend to have a structural "collapse", or rapid change in connectivity, as the tasks switch, consistent with the rapid change (and eventual increase) in validation loss.

Further, the frequency of neighborhood changes for hidden units throughout training (appearing as a crossing of unit trajectories in the visualization) corresponds to an increase in validation loss; this is due to a change in function of the hidden units, corrupting the intended use of such units for earlier

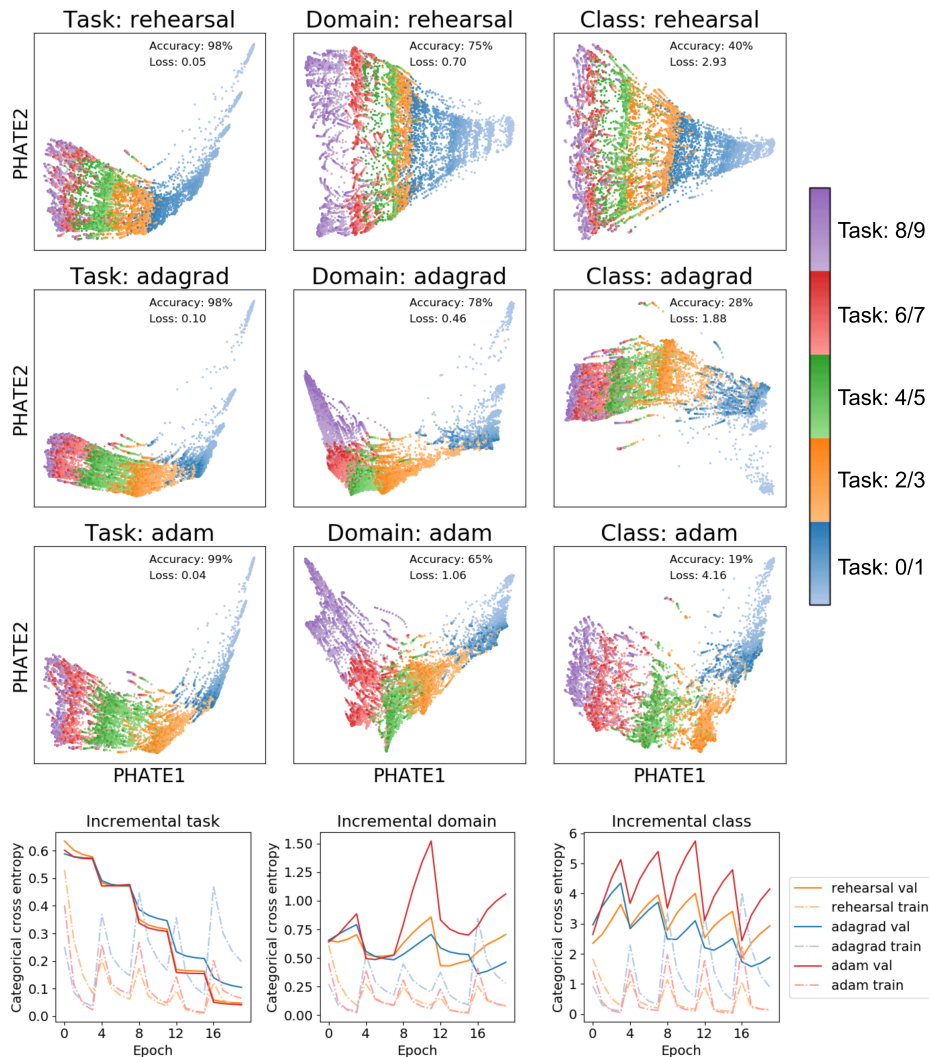

Figure 3: Visualization of a 2 layer MLP trained on Split MNIST for five-task continual learning of binary classification. Training loss and accuracy are reported on the current task. Validation loss and accuracy is reported on a test set consisting of an even number of samples from all tasks. Only 100 neurons are shown for clarity. Full plots are available in Section S4.

tasks. We quantify this effect by calculating the Adjusted Rand Index (ARI, Santos and Embrechts [32]) on cluster assignments computed on the subset of the visualization corresponding to the hidden units pre- and post-task switch, and find that the average ARI is strongly negatively correlated with the network's final validation loss averaged over all tasks ($\rho = 0.94$). Results are similar for the same experiment run in CIFAR10 ($\rho = 0.86$, see Section S4).

Looking for such signatures, including rapid changes in hidden unit structure and crossing of unit trajectories, can thus be used to understand the efficiency of continual learning architectures.

## 4.4 Generalization

Despite being massively overparametrized, neural networks frequently exhibit astounding generalization performance [33, 34]. Recent work has showed that, despite having the capacity to memorize, neural networks tend to learn abstract, generalizable features rather than memorizing each example, and that this behaviour is qualitatively different in gradient descent compared to memorization [35].

Table 2: Adjusted Rand Index of cluster assignments computed on the subset of the PHATE visualization corresponding to the hidden units pre- and post-task switch. ARI is averaged across all four task switches, 6 different choices of clustering parameter (between 3–8 clusters) and 20 random seeds. Loss refers to average validation loss averaged over all tasks after completion of training.

|  | Task | | | Domain | | | Class | | |
|  | Rehears. | Adagr. | Adam | Rehears. | Adagr. | Adam | Rehears. | Adagr. | Adam |
|---|---|---|---|---|---|---|---|---|---|
| Val. Loss | 0.047 | 0.104 | 0.042 | 0.709 | 0.462 | 1.062 | 2.904 | 1.884 | 4.156 |
| ARI | 0.741 | 0.772 | 0.716 | 0.719 | 0.768 | 0.740 | 0.614 | 0.632 | 0.466 |

Table 3: Summed variance per epoch of the PHATE visualization is associated with the difference between a network that is memorizing and a network that is generalizing. Memorization error refers to the difference between train loss and validation loss.

|  |  | Kernel | | | Activity | | Random | |
|  | Dropout | L1 | L2 | Vanilla | L1 | L2 | Labels | Pixels |
|---|---|---|---|---|---|---|---|---|
| Memorization | -0.09 | 0.02 | 0.03 | 0.04 | 0.11 | 0.12 | 0.15 | 0.92 |
| Variance | 382 | 141 | 50 | 46 | 0.47 | 0.15 | 0.42 | 0.03 |

In order to demonstrate the difference between networks that learn to generalize and networks that learn to memorize, we train a 3-layer MLP with 128 hidden units in each layer to classify MNIST with: no regularization; $L_1/L_2$ weight regularization; $L_1/L_2$ activity regularization; and dropout. Additionally, we train the same network to classify MNIST with random labels, as well as to classify images with randomly valued pixels, such networks being examples of pure memorization. Each network was trained for 300 epochs, and the discrepancy between train and validation loss reported.

We note that in Figure 4, the networks with the poorest generalization (i.e. those with greatest divergence between train and validation loss), especially Activity $L_1$ and Activity $L_2$, display less heterogeneity in the visualization. To quantify this, we calculate the sum of the variance for all time slices of each embedding and regress this against the *memorization error* of each network, defined as the discrepancy between train and test loss after 300 epochs (Table 3), achieving a Spearman correlation of $\rho = -0.98$. Results are similar for the same experiment run in CIFAR10 ($\rho = -0.97$, see Section S5).

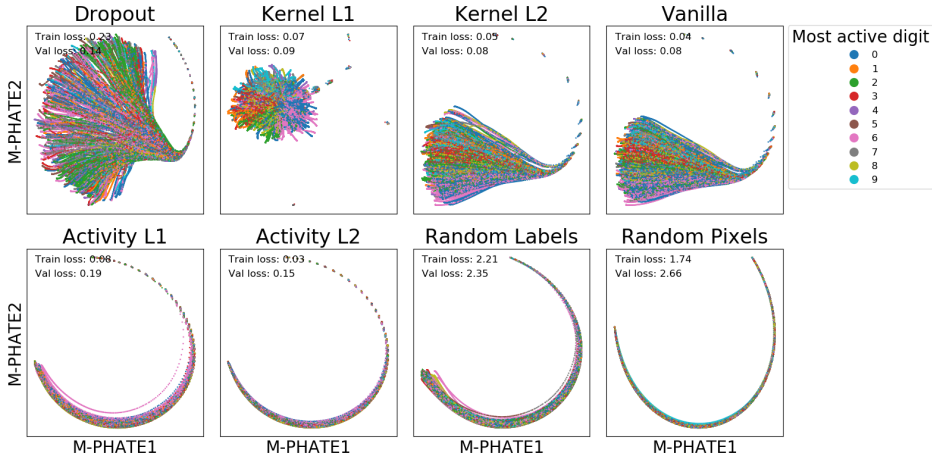

Figure 4: Visualization of a 3-layer MLP trained to classify MNIST with different regularizations or manipulations applied to affect generalization performance.

To understand this phenomenon, we consider the random labels network. In order to memorize random labels, the neural network must hone in on minute differences between images of the same true class in order to classify them differently. Since most images won't satisfy such specific

criteria most nodes will not respond to any given image, leading to low activation heterogeneity and high similarities between hidden units. The M-PHATE visualization clearly exposes this intuition visually, depicting very little difference between these hidden units. Similar intuition can be drawn from the random pixels network, in which the difference between images is purely random. We hypothesize that applying $L_1$ or $L_2$ regularization over the activations has a qualitatively similar effect; reducing the variability in activations and effectively over-emphasizing small differences in the hidden representation. This behavior effectively mimics the effects of memorization.

On the other hand, we consider the dropout network, which displays the greatest heterogeneity. Initial intuition evoked the idea that dropout emulates an ensemble method within a single network; by randomly removing units from the network during training, the network learns to combine the output of many sub-networks, each of which is capable of correctly classifying the input Srivastava et al. [36]. M-PHATE visualization of training with dropout recommends a more mechanistic version of this intuition: dropped-out nodes are protected from receiving the exact same gradient signals and diverge to a more expressive representation. The resulting heterogeneity in the network reduces the reliance on small differences between training examples and heightens the network's capacity to generalize. This intuition falls in line with other theoretical explorations, such as viewing dropout as a form of Bayesian regularization [37] or stochastic gradient descent [38] and reinforces our understanding of why dropout induces generalization.

We note that while this experiment uses validation data as input to M-PHATE, we have repeated this experiment in Section S2 and show equivalent results. In doing so, we provide a mechanism to understand the generalization performance of a network without requiring access to validation data.

## 5    Conclusion

Here we have introduced a novel approach to examining the process of learning in deep neural networks through a visualization algorithm we call M-PHATE. M-PHATE takes advantage of the dynamic nature of the hidden unit activations over the course of training to provide an interpretable visualization otherwise unattainable with standard visualizations. We demonstrate M-PHATE with two vignettes in continual learning and generalization, drawing conclusions that are not apparent without such a visualization, and providing insight into the performance of networks without necessarily requiring access to validation data. In doing so, we demonstrate the utility of such a visualization to the deep learning practitioner.

### Acknowledgments

This work was partially supported by the Gruber Foundation *[S.G.]*; the Chan-Zuckerberg Initiative (grant ID: 182702) and the National Institute of General Medical Sciences of the National Institutes of Health (grant ID: R01GM130847) *[S.K.]*; and the National Institute of Neurological Disorders and Stroke of the National Institutes of Health (grant ID: R01EB026936) *[G.M.]*.

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
