[Supplementary Material]

# Appendix for Visualizing the PHATE of Neural Networks

**Scott Gigante**
Computational Biology and Bioinformatics Program
Yale University
New Haven, CT 06511
scott.gigante@yale.edu

**Adam S. Charles**
Princeton Neuroscience Institute
Princeton University
Princeton, NJ, 08544
adamsc@princeton.edu

**Smita Krishnaswamy**
Departments of Genetics and Computer Science
Yale University
New Haven, CT 06520
smita.krishnaswamy@yale.edu

**Gal Mishne**
Halıcıoğlu Data Science Institute
University of California, San Diego
La Jolla, CA 92093
gmishne@ucsd.edu

## S1    Multislice graph construction

In Section 3, we describe a multislice affinity kernel $K$ built from an *intraslice* kernel, which connects hidden units in the same epoch, and an *interslice* kernel, which connects each hidden unit to itself at different epochs. We further clarify the intuition behind such an affinity kernel in two schematics.

Figure S1 displays a graph of 10 hidden units in a dynamically changing graph structure over the course of four time slices. Each hidden unit's local neighborhood within its own time slice (its intraslice affinities) changes as the system evolves, with connectivity shown as black lines. Additionally, each hidden unit is connected to itself across different epochs, with strength of these interslice connections (shown as dotted lines) also dependent on similarities (rather than simply a fixed-weight connection).

Figure S2 displays the top left corner of an example of a multislice affinity kernel. The full multislice kernel ($\mathbf{K}((\tau, i), (\upsilon, j))$, left) is composed on the intraslice kernels placed down the block diagonal ($\mathbf{K}_{\text{intraslice}}^{(1)}(i, j), \ldots, \mathbf{K}_{\text{intraslice}}^{(\tau)}(i, j)$, middle) and the interslice kernels forming the diagonals of each off-diagonal block ($K_{\text{interslice}}^{(1)}(\tau, \upsilon), \ldots, K_{\text{interslice}}^{(i)}(\tau, \upsilon)$, right).

## S2    Selection of representative subset $Y$

In Section 3, we state that the representative subset $Y$ is taken from points not used in training. However, there is no reason why this should be the case. To demonstrate that M-PHATE can be used successfully without accessing data external to the training set, we show in Figure S3 a repetition of the generalization experiment, using only training data to build the visualization. Using the same quantification of variance and memorization as in Section 4.4, we obtain an equally strong correlation (Spearman's $\rho = -0.95$, Table S1). Further, we note that the visualizations are qualitatively very similar to those obtained using training data, indicating that M-PHATE can be used to understand the generalization performance of a network without having access to an external validation set.

## S3    Parameters for visualization methods comparison

In Section 4.2, we compare M-PHATE to Diffusion Maps, t-SNE and Isomap in both a standard and multiscale context. Since t-SNE and Isomap require distance matrices, not affinity matrices, we

Figure S1: Example schematic of the multislice graph used in M-PHATE. The intra- and interslice kernels represent the similarities between the graph nodes at different time-points, providing PHATE with a time-aware distance to visualize the data with.

Table S1: Summed variance per epoch of the PHATE visualization is associated with the difference between a network that is memorizing and a network that is generalizing, where the visualization is built using only training data. Memorization error refers to the difference between train loss and validation loss.

|  |  | Kernel | | | Activity | | Random | |
| --- | --- | --- | --- | --- | --- | --- | --- | --- |
|  | Dropout | L1 | L2 | Vanilla | L1 | L2 | Labels | Pixels |
| Memorization | -0.09 | 0.02 | 0.04 | 0.05 | 0.10 | 0.12 | 0.13 | 0.53 |
| Variance | 59 | 77 | 35 | 28 | 0.66 | 0.34 | 0.37 | 0.03 |

convert the multislice kernel to geodesic distances by computing the shortest-path over the graph with the distance $D = -\log K'$. For standard application of Isomap and t-SNE, we use the default parameters in `sklearn` [1]. Since diffusion maps can be applied to any symmetric non-negative affinity kernel and does not have a reference implementation, we apply diffusion maps to the adaptive bandwidth kernel built in PHATE.

## S4  Continual Learning

### Continual Learning Schemes

Hsu et al. [2] describe three schemes of continual learning commonly used in the literature.

$$\mathbf{K}((\tau, i), (\upsilon, j)) \qquad \mathbf{K}^{(\tau)}_{\text{intraslice}}(i, j) \qquad \mathbf{K}^{(i)}_{\text{interslice}}(\tau, \upsilon)$$

Figure S2: Example schematic of the multislice kernel used in M-PHATE. This kernel is a sum of intaslice and interslice affinities.

Figure S3: Visualization of a 3-layer MLP trained to classify MNIST with different regularizations or manipulations applied to affect generalization performance, where the visualization is built using only training data.

Incremental *task* learning describes the process of learning shared hidden units for separated output layers for each task; the output units for task $i$ are therefore protected from gradient signals during the training of task $j \neq i$. This is akin to the standard model of transfer learning, in which all but the final layer of a network are copied for a new task, with a fresh output layer attached for the new task.

Incremental *domain* learning describes the process of learning an entirely shared network which learns to perform all tasks separately, but with the same units; in this case the output units for task $i$ are the same units that are used in task $j$ and must learn to correctly classify training examples from separate tasks as though they were the same class.

Incremental *class* learning describes the process of learning an entirely shared network which learns to perform all tasks at once, with no knowledge of which task is currently being performed. The network contains separate output units for each task, but must select which output units to use, in contrast to incremental task learning in which the task is specified. This is by far the most difficult setting, since in training any one task, the optimal solution is to never predict the output classes of any other task; this strongly encourages catastrophic forgetting.

Figure S4 demonstrates these three architectures on Split MNIST.

Figure S4: Architectures for incremental learning scenarios. Reproduced with permission from Hsu et al. [2].

## Network Parameters

The networks in Section 4.3 are trained as follows. Input data is scaled from 0 to 1. All networks consist of a MLP with 2 layers of 400 units with ReLU activation, and a softmax classification output layer. All networks are trained with a batch size of 128, split to batches of 64 new data and 64 rehearsal data in the case of Naive Rehearsal. For the Adam optimizer, we use a learning rate of $1e^{-5}$. For the Adagrad optimizer, we use a learning rate of $1e^{-4}$. For Naive Rehearsal, we use the Adam optimizer. All networks are built and trained in Keras using a Tensorflow backend.

## MNIST Full Results

Figure 3 shows the visualizations of the continual learning networks for a subset of 100 hidden units from each layer of the MLP with 2 layers of 400 units. Figures S5 and S6 show the full embedding of layers 1 and 2 respectively. In all cases, the visualizations are computed on all hidden units and subsampled for plotting purposes only.

We note the striking difference between layer 1 and layer 2 in all visualizations. In each case, there is a strong vertical pattern in layer 2, indicating that layer 2 is undergoing very large changes in hidden representation such that successive time-slices of the network are largely disconnected from one another. This can be most clearly seen in ADAM Incremental Class learning, in which the network appears to entirely forget the learned representations in layer 2, which is corroborated by the validation loss, which resets to the same point after each task. In comparison, the Naive Rehearsal visualizations remain connected at each task switch, which is consistent with the improved capacity of the network to retain the performance achieved on previous tasks.

## CIFAR10 Results

To show that the results shown above generalize beyond one specific dataset, we repeated the same experiment with CIFAR10. Since this continual learning task is substantially more difficult with CIFAR10 than with MNIST, we doubled the number of epochs per task to 8. The layer-wise M-PHATE embeddings of the learning process are shown in Figures S7 and S8. As with MNIST, we see that the simpler tasks retain more structure in the visualization at the point of task switch. Additionally, the vertical patterning observed in layer 2 is seen here once again, indicating that this is a feature of the network, rather than the task.

Figure S5: Visualization of layer 1 of a 2 layer MLP trained on Split MNIST for five-task continual learning of binary classification. Accuracy is reported on a test set consisting of an even number of samples from all tasks.

Table S2: Adjusted Rand Index of cluster assignments computed on the subset of the PHATE visualization corresponding to the hidden units pre- and post-task switch on networks trained on Split CIFAR10 for 8 epochs on each task. ARI is averaged across all four task switches, 6 different choices of clustering parameter (between 3–8 clusters) and 20 random seeds. Loss refers to average validation loss averaged over all tasks after completion of training.

| | Task | | | Domain | | | Class | | |
|---|---|---|---|---|---|---|---|---|---|
| | Rehears. | Adagr. | Adam | Rehears. | Adagr. | Adam | Rehears. | Adagr. | Adam |
| Val. Loss | 0.483 | 0.673 | 0.675 | 0.568 | 0.644 | 0.723 | 5.403 | 6.724 | 8.407 |
| ARI | 0.478 | 0.571 | 0.372 | 0.541 | 0.605 | 0.356 | 0.265 | 0.259 | 0.140 |

Once again, we quantify the effect of structural collapse in the visualization by calculating the Adjusted Rand Index (ARI) on cluster assignments computed on the subset of the visualization corresponding to the hidden units pre- and post-task switch, and find that the average ARI is strongly negatively correlated with the network's final validation loss averaged over all tasks ($\rho = 0.86$, Table S3), as it was with the MNIST experiment.

Figure S6: Visualization of layer 2 of a 2 layer MLP trained on Split MNIST for five-task continual learning of binary classification. Accuracy is reported on a test set consisting of an even number of samples from all tasks.

## S5   Generalization

**Network Parameters**

The networks in Section 4.4 are trained as follows. Input data is scaled from 0 to 1. All networks consist of a MLP with 3 layers of 128 units with Leaky ReLU activation with $\alpha = 0.1$, and a softmax classification output layer. All networks are trained with a batch size of 256 with the Adam optimizer and a learning rate of $1e^{-5}$. All regularizations are applied with a weight of $1e^{-4}$. Dropout is applied with $p = 0.5$. For the random labels network, we randomly permute the output labels of the training data, leaving the validation data intact. For the random pixels network, we randomly assign all pixel values from a standard normal distribution. All networks are built and trained in Keras [3] using a Tensorflow [4] backend.

**CIFAR10 Results**

To show that the results shown above generalize beyond one specific dataset, we repeated the same experiment with CIFAR10 (Figure S9). Once again, there is a strong association between entropy of the visualization and the final difference between training loss and validation loss ($\rho = -0.97$, Table S3), indicating that this representation of network generalization performance is a feature of the network, rather than the specific dataset used.

Figure S7: Visualization of layer 1 of a 2 layer MLP trained on Split CIFAR10 for five-task continual learning of binary classification. Training loss and accuracy are reported on the current task. Validation loss and accuracy is reported on a test set consisting of an even number of samples from all tasks.

Table S3: Summed variance per epoch of the PHATE visualization is associated with the difference between a network that is memorizing and a network that is generalizing when trained on CIFAR10. Memorization error refers to the difference between train loss and validation loss.

|  |  | Kernel | | | Activity | | Random | |
|---|---|---|---|---|---|---|---|---|
|  | Dropout | L1 | L2 | Vanilla | L1 | L2 | Labels | Pixels |
| Memorization | -0.12 | 0.12 | 0.15 | 0.15 | 0.38 | 0.46 | 0.23 | 8.71 |
| Variance | 74 | 61 | 42 | 28 | 0.45 | 0.18 | 0.41 | 0.02 |

## S6 M-PHATE parameters

All multislice graphs are built with $k = 2$, $\alpha = 5$ and $\kappa = 25$. We apply PHATE on the multislice affinity matrix with PHATE parameters $\gamma = 0$ and $n\_landmark = 3000$, and use the automatically selected parameter of $t$ provided by the PHATE algorithm.

## S7 Computing infrastructure

All computation was done on a single 36-core workstation running Arch Linux with a NVIDIA TITAN X graphics card and 512GB of RAM.

Figure S8: Visualization of layer 2 of a 2 layer MLP trained on Split CIFAR10 for five-task continual learning of binary classification. Training loss and accuracy are reported on the current task. Validation loss and accuracy is reported on a test set consisting of an even number of samples from all tasks.

## S8  Validation data selection

Training and validation data were separated into pre-defined groups as given in Keras [3].

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

Figure S9: Visualization of a 3-layer MLP trained to classify CIFAR10 with different regularizations or manipulations applied to affect generalization performance.