[Reviews · NeurIPS 2019]

Reviewer 1



Update after author response: Taking on faith the results the authors report in their author response (namely ability to identify generalization performance using only the training set, results on CIFAR10 and white noise datasets, and the quantitative evaluation of the task-switching), I would raise my score to a 6 (actually if they did achieve everything they claimed in the author response, I would be inclined to give it a 7, but I'd need to see all the results for that). ========== Originality: I think the originality is fairly high. Although the PHATE algorithm exists in the literature, the Multislice kernel is novel, and the idea of visualizing the learning dynamics of the hidden neurons to ascertain things like catastrophic forgetting or poor generalization is (to my knowledge) novel. Quality: I think the Experiments sections could be substantially improved: (1) For the experiments on continual learning, from looking at Figure 3 it is not obvious to me that Adagrad does better than Rehearsal for the "Domain" learning setting, or that Adagrad outperforms Adam at class learning. Adam apparently does the best at task learning, but again, I wouldn't have guessed from the trajectories. I would have been more convinced if there were a quantitative metric to support the claim that "the highest-performing networks all tend to preserve representational structure across changing tasks". (2) For the experiments on generalization, I will first note that the claim that "increased heterogeneity among hidden units correlate with improved generalization performance" is on the surface a bit counter-intuitive, because one might be tempted to associate "increased heterogeneity" with a more complex decision boundary, and more complex decision boundaries typically generalize worse. The authors provide an explanation, which is that "in order to memorize scrambled labels, the neural network must hone in on minute differences between images of the same true class in order to classify them differently. Since most images won’t satisfy such specific criteria most nodes will not respond to any given image, leading to low activation heterogeneity and high similarities between hidden units". This is an interesting claim, but it would be better supported if the authors explored datasets other than MNIST. This is because a network could also learn to "hone in on minute differences" by learning an extremely complex decision boundary that involves the firing of multiple neurons. In fact, part of the intuition for dropout is to discourage the kind of complex co-adapted firing of neurons that can lead to overfitting. MNIST images are predominantly black, and I am wondering if this is related to the fact that the network learns to memorize MNIST by keeping most neurons silent on most images. Does the same hold true for networks trained to memorize "white noise" images? Clarity: The clarity of the paper is good. It was an enjoyable read. Significance: The significance is intermediate. As mentioned, the claims in the Experiments section could use more support. More generally, I did not feel that the authors have made a strong case that M-PHATE can be used to draw conclusions that one would not anyway draw by looking at the validation set accuracy/loss. Minor comments: - There is a typo with "On the other had". - There seems to be a typo in line 111 where t was used instead of tau. - What is alpha in the formula for K_intraslice? - In line 113, the authors say epsilon is the fixed "intraslice" bandwidth - but epsilon appears in the formula for the interslice kernel, so did they mean to say interslice? - Line 123 mentions the kernel is symmetrized, though I am confused because the formulas for K_interslice and K_intraslice already seem symmetric as they involve taking the magnitude of differences. Can the authors clarify why the kernel is not already symmetric? - Please proofread line 162; the sentence is difficult to parse.

Reviewer 2



Overall, this is a well motivated, executed, and clearly written paper that extends the existing literature on visualizing NN evolution. The supplemental material helped clarify some of my questions about the experimental setup (e.g., the difference between various continual learning tasks). The solution direction is clearly formalized and technically sound. The authors use an appropriate experimental setup. However, the authors do not discuss the weaknesses of the approach and experiments or list future direction for readers. The writeup is exceptionally clear and well organized-- full marks! I have only minor feedback to improve clarity: 1. Add a few more sentences explaining the experimental setting for continual learning 2. In Fig 3, explain the correspondence between the learning curves and M-PHATE. Why do you want to want me to look at the learning curves? Does worse performing model always result in structural collapse? What is the accuracy number? For the last task? or average? 3. Make the captions more descriptive. It's annoying to have to search through the text for your interpretation of the figures, which is usually on a different page 4. Explain the scramble network better... 5. Fig 1, Are these the same plots, just colored differently? It would be nice to keep all three on the same scale (the left one seems condensed) M-PHATE results in significantly more interpretable visualization of evolution than previous work. It also preserves neighbors better (Question: why do you think t-SNE works better in two conditions? The difference is very small tho). On continual learning tasks, M-PHATE clearly distinguishes poor performing learning algorithms via a collapse. (See the question about this in 5. Improvement). The generalization vignette shows that the heterogeneity in M-PHATE output correlates with performance. I would really like to recommend a strong accept for this paper, but my major concern is that the vignettes focus on one dataset MNIST and one NN architecture MLP, which makes the experiments feel incomplete. The results and observations made by authors would be much more convincing if they could repeat these experiments for more datasets and NN architectures.

Reviewer 3



-- Post author-feedback comments -- I would like to thank the authors for their response to my concerns and for clarifications on the methods. The authors seem to have done much work to improve their submission. In particular, I am glad they have included another dataset to evaluate their algorithm. I still do think they should explore other choices for the kernel definition, in particular, the values they choose for $K((\tau, i), (\nu, j))$ when $\tau \ne \nu$ and $i \ne j$. There is a full spectrum of possibilities between regular PHATE and M-PHATE in terms of what the structure of $K$ can be. I do recognize it is a non-trivial choice, and perhaps useful to include in your future work. As such, I raised my score for the submission to 6. -- Initial review -- The main contribution of this paper is an extension to the PHATE algorithm by Moon et al. [14] -- here coined: the Multislice PHATE (M-PHATE). The algorithm relies heavily on a proposed a multislice kernel that quantifies similarities between different units of a NN within the same epoch as well as between the same unit but at different training timepoints. Originality: ------------ M-PHATE slightly modifies PHATE by defining a heuristic multislice kernel over all the hidden units and all epochs of training. The proposed kernel is composed of a varying-bandwidth Gaussian kernel for entries corresponding to different units for at the same epoch (intraslice), and a fixed-bandwidth Gaussian kernel for same units across different epochs (interslice), zero value is assigned to all other entries. Gaussian kernels are a standard choice for quantifying pairwise affinities; the paper justifies the selection of the fixed bandwidth for the interslice part, but provides no reasoning as to why adaptive bandwidths should be used for the intraslice part. The new coordinates for visualization are computed following the same procedure as in the existing PHATE [14] based on MDS over the information distance ($\|\log P^t(i) - \log P^t(j)\|$) defined for the row-normalized multislice kernel ($P = D^{-1}K$). Novelty of this paper lies more in new applications than new technical methodology. The authors use M-PHATE to visualize the training process with different regularization techniques or the continual learning task with different optimization training schemes. Quality: -------- M-PHATE uses k-th nearest neighbor adaptive bandwidth for the Gaussian kernel, which can be very unstable and sensitive to outliers. Perhaps an average over the first 1 to k nearest neighbors, or an entropy equalizing based procedure (like the one used in t-SNE) is a more robust strategy. Apart from that, using the varying bandwidth means that method stretches and contracts different unit neighborhoods at different rates. It is not clear why the authors utilized a fixed bandwidth kernel for temporal" (iterslice) but not for spatial" (intraslice) variation between units. There can also be interactions between neighboring units at different epochs, and assigning zero entries corresponding to these combinations ($K((\tau, i), (\nu, j))$ for $\tau \ne \nu$ and $i \ne j$) night not be adequate. With the current choice of the multislice kernel, with no interactions allowed between different units across different epochs, it is somewhat unsurprising that the resulting visualizations mainly show temporal changes and unit trajectories over training time. The experiments included in the paper were reasonable and helpful for understanding the objectives of the paper. However, the quantitative as well as visual results do not conclusively show M-PHATE's superiority over other methods. The effectiveness of visualizations was evaluated only with the neighbor preservation, which cannot capture the quality large scale or global structures representation. M-PHATE also does not outperform t-SNE at inteslice neighbor-preservation, which is concerning as the method was intended for displaying the NN dynamics. The parameter selection for t-SNE was not reported, raising a question of whether t-SNE would performed even better if e.g. perplexity setting was adequately tuned. The visual results show that M-PHATE seems to produce visualizations that have a more coherent global structure consistent with the training evolution over epochs. However, it seems that the M-PHATE show only the dynamics, and the intraslice variation and organization of the units within the same epoch cannot be easily perceived due to crowding of the datapoints. Clarity ------- Overall, the paper is clearly written, and it's easy to understand the ideas proposed. However, there are some parts that require clarification, e.g. the paper does not explicitly state what is the standard kernel''. I assumed that the kernel was simply a Gaussian kernel over the z-score activations, but it is not clear if adaptive bandwidth or fixed were used. Minor typo: line 111 I think should read (...) j-th unit at epoch $\tau$ (...)'', not (...) j-th unit at epoch t (...)''. Significance ------------- Developing visualization tools to improve the understanding of the behavioud of the NNs throughout the training process is important, and and the authors of this paper made a good effort to tackle this task. Even though, the choices in the paper might have not been optimal or fully tested and explored, the paper makes a step forward by explaining how hidden units can be embedded, if a suitable similarity kernel can be defined between units and different training epochs. I liked the idea of comparing different training optimization schemes (here Adagrad, Adam, Naive Rehearsal) applied to continual learning through visualization. It was also useful to see visualization of NNs trained with different regularization schemes, exhibiting different patterns on the M-PHATE plot, depending on generalization performance ranging on a spectrum from pure memorization to high generalization. Although, it seems the interpretation of M-PHATE plots is still not straightforward and should be caustiously considered.

[Author Response · NeurIPS 2019]

We thank the reviewers for their time and the many positive comments, in particular their appreciation of the novelty, originality, and clarity of our work. As a result of the thoughtful reviews, we have strengthened our work by 1) Re-running all quantification on training data in addition to validation data, showing that M-PHATE can evaluate performance without validation data; 2) Showing via running M-PHATE on white noise that generalization results are not purely due to MNIST being predominantly zeros; 3) Defining the results of the task-switch experiment quantitatively; and 4) Repeating experiments on CIFAR10, showing that our results are robust. Specific points below will be incorporated into the manuscript.

**Validation set (R1)**: The new runs on training data show that our method identifies generalizability without requiring access to validation data. As opposed to differences in loss/accuracy, the dynamics of units revealed through M-PHATE retain key geometric features of the network when evaluated with both training and validation data. Thus M-PHATE is applicable even when a validation loss cannot be computed. We applied this to the generalization experiment using only training data and obtained $\rho = 0.93$, consistent with prior results.

**Other datasets (R1,R2,R3)**: We added a generalization experiment on both CIFAR10, ($\rho = 0.95$; consistent with MNIST results) and white noise (finding homogeneous structure similar to scrambled classes, see Fig. 1b).

**Quantitative evaluation of task-switch (R1)**: We added a metric to measure loss of structure by computing Adjusted Rand Index averaged over clusterings of the visualized units pre- and post-task switch (4 task switches $\times$ 6 parameters (3–8 clusters) $\times$ 20 repetitions). High structure preservation is strongly associated with low validation loss ($\rho = 0.90$).

**Weaknesses and future directions (R2,R3)**: *Other architectures*: Extending M-PHATE to CNNs/RNNs is a very interesting future direction. Defining appropriate similarity measures to enable comparison of units across layers for these networks will require non-trivial extensions beyond the scope of this work. Also, using M-PHATE in its current form on such networks will be slow due to the $O(n^2)$ complexity. We plan to explore such extensions, both in designing appropriate metrics and developing computationally efficient models (e.g., online computation). *Overcrowding*: There is always information loss in low-dimensional visualizations of all units at all time points, especially in large networks over many epochs. Despite this, we show in specific tasks that local structure of the units can still be informative.

**Higher dimensions (R3)**: We will add supplementary figures and quantitative measures for 2D vs 3D embeddings. Beyond 3D, dimensionality reduction methods become difficult to visually interpret (the initial intended use).

**User studies (R2)**: We agree feedback is invaluable and will publish our code to get such feedback from the community.

**Kernel details (R3)**: The "standard" kernel differs for each algorithm: t-SNE uses the perplexity kernel, ISOMAP the k-NN kernel, and PHATE and DM use the adaptive bandwidth Gaussian kernel (defined in the paper).

**t-SNE/neighborhood preservation (R2, R3)**: It is well known that t-SNE does not preserve global structure, and that PHATE does [Moon et al. 2017, Linderman and Steinerberger 2019]. Here, preserving interslice vs. intraslice neighbors has an inherent trade-off. t-SNE guarantees almost perfect interslice neighborhood preservation by clustering each hidden unit with only itself, at the cost of losing all of the intraslice neighbors. We further show t-SNE's lack of utility in Figure 1a, in which you see no useful difference in the embeddings between the dropout and scrambled networks from the generalization experiment; M-PHATE showed significant, obvious, and interpretable differences. t-SNE (as all other methods) was run with default parameters in scikit-learn.

**Visualization quality metrics (R3)**: Since standard algorithms do not provide any useful visualization of the data (see Fig 1a as an example), an algorithm that faithfully represents the structure of the raw data (e.g. as measured by coranking) is not expected to perform well. We will extend the notion of coranking to interslice and intraslice neighborhoods and report the result of this to give a notion of global structure in the multislice context.

**Neighboring unit interactions (R3)**: Kernel construction over all node similarities and ignoring *a priori* knowledge of the data yields standard PHATE (paper, Figure 2), which is a poor visualization of the data. In addition, the construction of the multislice kernel such that $K((\tau, i), (\nu, j)) = 0$ for $\tau \neq \nu, i \neq j$ is standard in prior work in multislice graph construction (Mucha et al., 2010).

**Deeper insights/conclusions from M-PHATE (R1,R2)**: The collapse in Adam is likely due to the extremely large gradients causing a rapid build-up of momentum and pushing all units in the same direction much farther than their current spread. The lack of heterogeneity of the activity regularizers is a direct consequence of their formulation: forcing all activities to be small induces similar activations. For continual learning, in appendix Fig. S4, S5 we show separate plots for the layers of the task switch network. These visualizations depict that the significant changes to the network due to the task switch are happening in layer 2, which seems to undergo rapid learning after each task change and then stays static. Layer 1 is highly heterogeneous, indicating a more universal representation which is evolving more smoothly throughout training when compared to layer 2. These layer-specific insights are not accessible from plotting global validation metrics.

Figure 1: A) tSNE on FFNNs with (left) scrambled labels (right) dropout; B) M-PHATE on an FFNN trained on white noise.

[Meta-Review · NeurIPS 2019]

The reviewers are all positive if not wildly so, and as the response suggests I would like to not put too much weight on the specific scores. This is a good submission that has a small number of clearly defined improvements outlined in the extensive and helpful reviews.